# Drug Therapies for Diabetes

**DOI:** 10.3390/ijms242417147

**Published:** 2023-12-05

**Authors:** Roni Weinberg Sibony, Omri Segev, Saar Dor, Itamar Raz

**Affiliations:** 1Faculty of Medicine, Ben-Gurion University, Beer Sheva 8443944, Israel; roniweinb@gmail.com (R.W.S.); saardorh@gmail.com (S.D.); 2Faculty of Medicine, Tel Aviv University, Tel Aviv 69978, Israel; omrihsegev@gmail.com; 3Faculty of Medicine, Hebrew University of Jerusalem, Jerusalem 9112001, Israel; 4Diabetes Unit, Department of Endocrinology and Metabolism, Hadassah Medical Center, Jerusalem 91240, Israel

**Keywords:** T2DM, disease management, combination therapy, SGLT-2I, GLP1 RA

## Abstract

The treatment of type 2 diabetes (T2D) necessitates a multifaceted approach that combines behavioral and pharmacological interventions to mitigate complications and sustain a high quality of life. Treatment encompasses the management of glucose levels, weight, cardiovascular risk factors, comorbidities, and associated complications through medication and lifestyle adjustments. Metformin, a standard in diabetes management, continues to serve as the primary, first-line oral treatment across all age groups due to its efficacy, versatility in combination therapy, and cost-effectiveness. Glucagon-like peptide-1 receptor agonists (GLP-1 RA) offer notable benefits for HbA1c and weight reduction, with significant cardiovascular benefits. Sodium-glucose cotransporter inhibitors (SGLT-2i) lower glucose levels independently of insulin while conferring notable benefits for cardiovascular, renal, and heart-failure outcomes. Combined therapies emphasizing early and sustained glycemic control are promising options for diabetes management. As insulin therapy remains pivotal, metformin and non-insulin agents such as GLP-1 RA and SGLT-2i offer compelling options. Notably, exciting novel treatments like the dual GLP-1/ glucose-dependent insulinotropic polypeptide (GIP) agonist show promise for substantially reducing glycated hemoglobin and body weight. This comprehensive review highlights the evolving landscape of pharmacotherapy in diabetes, the drugs currently available for treating diabetes, their effectiveness and efficacy, the impact on target organs, and side effects. This work also provides insights that can support the customization of treatment strategies.

## 1. Introduction

The last decade has seen the advent of new medications for lowering blood glucose levels. These medications also exhibit a remarkable capacity to alleviate cardiovascular risk factors. These drugs demonstrate a significant ability to reduce the rates of cardio-renal events in patients with diabetes, as well as in individuals without the disease [1].

These medications have shifted the focus of cardiologists, nephrologists, and healthcare professionals towards preventing cardio-renal events, partially at the expense of focusing on the importance of maintaining near-normal blood glucose levels [1,2,3]. Simultaneously, several influential studies have highlighted the crucial role of maintaining optimal blood glucose levels for preventing both microvascular and macrovascular complications. These studies raise the fundamental question of whether a 7% hemoglobin A1c (HbA1c) level is an adequate goal or whether we should strive for the normalization of blood glucose levels [4,5].

These studies have also prompted discussions about the early administration of drugs in patients with diabetes, potentially as combination therapies, and their possible role in the management of diabetes and newly diagnosed diabetes [6]. The introduction of these new medications has led to inquiries regarding the role of older drugs in diabetes management and whether discontinuing some of them may be warranted due to the associated potential for harm [5,7].

This review provides a comprehensive overview of oral and injectable non-insulin medications used to improve blood glucose levels (Table 1). The results underscore the importance of early diagnosis and early treatment with combination therapy [4,7,8]. We delve into the concept of precision medicine and explore the role of older drugs in patient management. Furthermore, we present a proposal for a groundbreaking approach to the treatment of type 2 diabetes, with an emphasis on striving for a cure for the disease.

## 2. Metformin

Sixty years after it was introduced into the diabetes pharmacopeia, metformin remains a cornerstone in the treatment of type 2 diabetes and is recommended as the primary oral drug of choice for the management of T2DM across all age groups [9]. The well-known advantages of this agent include its glucose-lowering efficacy, ease of combination with almost any other glucose-lowering agent, and its low cost. Metformin is well-tolerated, has only mild side effects, carries a low risk of hypoglycemia, and provides modest body-weight reduction [10,11].

Metformin, the first-line medication for treating type 2 diabetes, has complex and not yet fully elucidated mechanisms of action that extend beyond its traditionally understood effects on glucose regulation in the liver. While early evidence highlighted the liver as a primary site of action, recent studies indicate a more multifaceted process involving other body areas, such as the gastrointestinal tract, gut microbiota, and tissue-resident immune cells. Its effects seem to differ based on dosage and treatment duration. While the drug was initially thought to primarily affect hepatic mitochondria, new research suggests a novel target at low concentrations, possibly at the lysosome surface. This research suggests a new mode of action. Amidst ongoing debates and the complexity of available information, recent discoveries suggest that the impact of metformin extends beyond liver functions, involving the gut microbiota and immune-system modulation. The drug’s effects could thus potentially expand its use to conditions beyond diabetes [12].

A recently published meta-analysis suggested that metformin may protect the cardiovascular system. It also noted that not only could the drug be used more widely to improve kidney function, but that it could contribute to kidney protection. Data also indicate that metformin may reduce the risk of neurodegenerative conditions, and trials are ongoing to directly evaluate the drug’s anti-neoplastic properties [13]. Regarding dementia, a meta-analysis demonstrated that diabetic patients treated with metformin had a significantly lower prevalence of cognitive impairment (odds ratio = 0.55, 95% CI 0.38 to 0.78). Furthermore, the incidence of dementia was also significantly reduced in this group (hazard ratio = 0.76, 95% CI 0.39 to 0.88) [14].

Regarding the anti-neoplastic properties of biguanides, metformin has emerged as a promising candidate for anticancer treatment. Accumulating epidemiological, preclinical, and clinical evidence have provided support for the utilization of metformin as a therapeutic agent in cancer [15,16]. There are over 50 recent or active clinical trials investigating the use of metformin for human malignancies. Of particular significance is its ability to reduce circulating insulin levels, which may have significant implications for the treatment of malignancies associated with hyperinsulinemia, such as breast and colon cancers. Furthermore, metformin may exert direct inhibitory effects on cancer cells by targeting the signaling pathway of mammalian target of rapamycin (mTOR) and interfering with protein synthesis [16]. Two studies investigated the effects of metformin administered over periods of three to six months to women who had completed chemotherapy and radiation treatment for breast cancer. The first study focused on women with plasma insulin levels of at least 45 pmol/L. This selection criterion was based on earlier research that had identified these women as being at a heightened risk for breast cancer. In this study, metformin was shown to reduce circulating insulin levels by 22.4% (*p* = 0.024) [17].

The second study involved women with elevated testosterone levels and compared metformin doses of 1000 mg/day and 1500 mg/day. The higher dose was found to significantly lower serum testosterone levels and the free androgen index compared to the lower dose. These findings highlight the potential of metformin to reduce serum markers associated with an increased risk of breast cancer [17].

In terms of the impact of metformin on cardiovascular disease, a meta-analysis of 13 trials (which included 2079 individuals with T2D who were allocated to metformin and a similar number to the comparison groups of diet and lifestyle changes or placebo) yielded favorable results for the prevention of myocardial infarction/ischemic heart disease. However, no effect reached statistical significance, mainly due to the absence of high-quality evidence [18]. The “UKPDS” study included 4075 early diagnosed diabetic patients randomly assigned to either a healthy-lifestyle intervention or healthy-lifestyle intervention with ADD. Among 342 obese diabetic patients that was randomized to metformin therapy, there was relative risk reduction (RRR) of −32% for diabetes-related endpoints, −42% for diabetes-related deaths, −39% for myocardial infarction, and −36% for overall mortality [19].

Furthermore, a recent clinical trial presented promising findings regarding long Corona virus disease (COVID). Outpatient treatment with metformin reduced the incidence of long COVID by 40% compared with placebo [20]. 

Despite extensive, long-standing experience with the clinical use of metformin, its mode of action is still not fully understood [9].

Notwithstanding the significant benefits of metformin, its high efficacy, and its low risk of side effects, there is increasing consensus that other approaches may be more appropriate as a first line of treatment for some patients [21].

## 3. Thiazolidinediones

Thiazolidinediones (TZD) work by binding to the peroxisome proliferator-activated receptor gamma (PPAR-γ) in the cell nucleus. This binding modulates gene expression involved in glucose and lipid metabolism, enhancing insulin sensitivity in muscle, fat, and liver cells. The primary advantages of TZD include improved glycemic control by reducing blood sugar levels, potential preservation of pancreatic beta cell function, and in some cases, positive impacts on lipid profiles, cardiovascular health, and the reduction of inflammation [22]. On the other hand, treatment with pioglitazone can increase the risk for weight gain, peripheral and retinal edema, dyspnea, hospitalization for heart failure (HHF), and bone fractures, mainly in women [23].

The PROactive (PROspective pioglitAzone Clinical Trial In macroVascular Events) trial presented promising data regarding the beneficial effects of pioglitazone. The main secondary endpoint was the composite of all-cause mortality, non-fatal myocardial infarction, and stroke. The results presented a significant decrease in fatal/nonfatal MI (excluding silent MI) [HR = 0.77; 95% CI 0.60–1.00; *p* = 0.046] and the composite of cardiovascular death, MI (excluding silent MI), and stroke (HR = 0.82; 95% CI 0.70–0.97; *p* = 0.020) [24].

In contrast, a meta-analysis published in 2022 that examined the effects of pioglitazone on cardiovascular events and all-cause mortality in patients with type 2 diabetes found that pioglitazone did not significantly affect major adverse cardiovascular events (MACE), all-cause mortality, or HHF (MH–OR: 0.90, 95% CI 0.78–1.03, 0.91, 95% CI 0.77–1.09) and (MH–OR: 1.16, 95% CI 0.73–1.83), respectively) [25]. The IRIS (Insulin Resistance Intervention after Stroke) trial was a multicenter, double-blind trial that randomly assigned 3876 patients who had recently suffered ischemic stroke or transient ischemic attack to receive either pioglitazone or a placebo. Among the patients who received pioglitazone, the risk of stroke or myocardial infarction was lower than it was among those who received the placebo (HR 0.76, 95% CI, 0.62–0.93; *p* = 0.007)) [23]. Pioglitazone has positive effects in nonalcoholic steatohepatitis (NASH) in both diabetic and non-diabetic patients [26]. In 2006, a randomized control trial with 55 patients presented proof of concept that pioglitazone plus diet, as compared with diet plus placebo, normalized liver aminotransferase levels, as it decreased plasma aspartate aminotransferase levels by 40% vs. 21% (*p* = 0.04), decreased alanine aminotransferase levels by 58% vs. 34%, (*p* < 0.001), decreased hepatic fat content by 54% vs. 0% (*p* < 0.001), and increased hepatic insulin sensitivity by 48% vs. 14% (*p* = 0.008) [27].

A meta-analysis that evaluated the effects of thiazolidinediones for the treatment of patients with prediabetes or T2DM combined with Nonalcoholic fatty liver disease (NAFLD) found that pioglitazone significantly improved insulin sensitivity and the results of liver histology. Additionally, it significantly reduced fasting blood glucose, glycosylated hemoglobin, plasma AST, ALT, and other liver biological indicators [28].

In 2012, the American Association for the Study of Liver Disease (AASLD) added pioglitazone as a treatment option for patients with biopsy-proven NASH [29].

## 4. Dipeptidyl Peptidase-4 Inhibitors

Dipeptidyl peptidase-4 inhibitors (DPP-4i) are common oral anti-hyperglycemic agents that are widely used worldwide. DPP-4 is an important modulator of the incretin system. DPP-4 inhibitors increase the concentrations of both active incretin hormones. They affect levels of GLP-1 by removing the N-terminal His7Ala8 from the active form of GLP-1 and glucose-dependent insulinotropic polypeptide [30]. Blocking incretin degradation with DPP4-i allows postprandial insulin release. The first DPP4-i, sitagliptin, received U.S Food and Drug Administration (FDA) approval in 2006. Extensive data from major clinical trials present the benefits of DPP4-i, such as lowering of HbA1c levels and reductions in inflammation and adipocyte size [30]. Some DPP4-i, like sitagliptin and saxagliptin, did not increase risk of major adverse cardiovascular events in diabetic patients with known cardiovascular disease [31,32]. In the SAVOR trial, saxagliptin increased the risk for HHF. Similar non-significant results were demonstrated in the EXAMINE trial [33], although the reason is currently unknown [32,34]. Therefore, the FDA does not recommend saxagliptin or alogliptin for patients with HF. DPP4-i may also decrease renal microalbuminuria. The SAVOR-TIMI53 trial presented positive renal outcomes with saxagliptin, which reduced the development and progression of microalbuminuria in patients with diabetes [32,35]. DPP4-i are known to be weight-neutral and carry a very low risk of hypoglycemia. Concerns were raised about possible risks for pancreatic cancer, neuroendocrine tumors, and pancreatitis with DPP4-i treatment. However, in 2014, the FDA and the European Medicines Agency (EMA) announced they could not establish a clear relationship between DPP4-i and pancreatitis or pancreatic cancer [36].

## 5. Sulfonylureas

Sulfonylureas lower blood glucose levels by increasing insulin secretion in the beta cells by blocking the KATP channels. They also limit gluconeogenesis in the liver. Sulfonylureas decrease the breakdown of lipids to fatty acids and reduce clearance of insulin in the liver [10]. 

Sulfonylureas are assessed as having high efficacy in lowering blood glucose levels but lack a durable effect and is associated with weight gain and hypoglycemia [10,21]. Use of sulfonylureas or insulin for early intensive blood glucose control significantly decreased the risk of microvascular complications, underscoring the importance of early and continued glycemic management [37].

At the same time, due to their glucose-independent stimulation of insulin secretion, sulfonylureas are associated with an increased risk of hypoglycemia [10]. Moreover, concerns about their cardiovascular safety have been raised in several retrospective studies, suggesting a greater risk of cardiovascular disease in patients treated with some of the sulfonylureas that prevent the preconditioning ischemia in the heart [38].

Nonetheless, sulfonylureas such as gliclazide and repaglinide do not block NA+/K+ ATPase in the heart vessels, and were not shown to increase cardiovascular events or mortality. [39].

Because novel glucose-lowering agents are available, because sulfonylureas carry the risk of hypoglycemia and weight gain, and because the effects of sulfonylureas decline over the long term, sulfonylureas should not be considered as first- or second line. Their use should be considered in patients who are not well-controlled with metformin, and for whom SGLT-2i, GLP1 RA, and DPP-4i are contraindicated or not tolerated, unavailable, or unaffordable. The expert opinion consensus panel published in Diabetes, Obesity and Metabolism in 2020 advised including sulfonylureas as a third-line treatment option [40]. A quadruple treatment combination therefore would include metformin, SGLT-2i, GLP-1RA, and sulfonylureas to provide additional decreases in HbA1c, mainly to achieve microvascular protection [40].

## 6. Glinides

Glinides, also known as meglitinides, are insulin secretagogues that depolarize pancreatic β-cells and consequently increase insulin release. Glinides include the drugs repaglinide and nateglinide [41]. Glinides and sulfonylureas differ in structure, yet both stimulate insulin secretion through distinct β-cell receptors [9]. When used as monotherapies, they exhibit similar clinical efficacy [42].

Glinides are rapid-acting, prandial glucose regulators that, unlike sulfonylureas, have a short duration of action. These characteristics render them effective in mitigating postprandial hyperglycemia when they are administered along with meals and are especially advantageous for individuals with inconsistent mealtimes [43]. Glinides are useful in patients with chronic kidney disease, as they are predominantly metabolized by the liver and therefore may be a useful alternative to metformin in this population [44].

In a Cochrane review of 15 trials that assessed the effects of glinides in patients with T2DM, both repaglinide and nateglinide led to reductions in HbA1c levels. Repaglinide resulted in a decrease of 0.1% to 2.1% in HbA1c, compared to 0.2% to 0.6% for nateglinide. Repaglinide was comparable to metformin in its effectiveness in reducing HbA1c levels, but nateglinide was not [45]. Combination therapy with metformin and glinides may improve glycemic control compared to metformin monotherapy [46]. 

Hypoglycemia is one of the major concerns associated with the use of glinides. Nevertheless, most episodes are mild to moderate and do not require assistance. Severe hypoglycemic episodes are rare but may occur; thus, glucose monitoring is important when treatment is initiated until safety is established [45,47]. Hypoglycemic events appear to be more common with sulfonylureas compared to glinides, and both can lead to weight gain [47].

Long-term studies investigating the effect of glinides on cardiovascular outcomes or mortality in patients with T2DM are lacking, and there is no compelling evidence that glinides increase cardiovascular risk [45,48]. Glinides are mostly used as an additional hypoglycemic agent for patients who fail to achieve their glycemic targets with metformin, but they are also useful as monotherapies when metformin or sulfonylureas are contraindicated.

## 7. Alpha-Glucosidase Inhibitors

Polysaccharides and disaccharides undergo enzymatic cleavage by alpha-glucosidase to monosaccharides in the upper intestine. Alpha-glucosidase inhibitors reversibly inhibit the enzymatic cleavage of complex carbohydrates to simple absorbable sugars, thereby reducing post-prandial hyperglycemia with no risk of hypoglycemia, subsequently reducing HbA1c [49].

In 1999, acarbose (Precose), an alpha-glucosidase inhibitor, was the first drug of this type approved by the FDA. In a multicenter, double-blind, placebo-controlled trial published in 1995, three doses of acarbose (100, 200, and 300 mg three times daily) were compared with placebo. After 16 weeks of treatment, acarbose-treated patients had significant reductions in mean HbA1c levels of 0.78%, 0.73%, and 1.10% (relative to placebo) in the 100 mg, 200 mg, and 300 mg groups, respectively. Gastrointestinal side effects (e.g., abdominal pain, flatulence, and diarrhea) and elevated serum transaminase levels were reported more frequently by the patients treated with acarbose than by those who received the placebo [50].

Acarbose can be an option for diabetic patients with constipation. Note that in the case of hypoglycemia in patients with concurrent treatment with acarbose, glucose/dextrose-only solutions are recommended.

## 8. Glucagon-like Peptide-1 Receptor Agonists—GLP 1 RA

GLP-1 RA has an excellent effect on the cardiovascular system and the brain, but its effects on heart failure and kidney failure are controversial. GLP-1 is a peptide hormone with multiple effects, including increasing insulin secretion and decreasing glucagon secretion in a glucose-dependent matter. Additionally, GLP-1 delays gastric emptying and increases satiety. Currently, there are multiple options for the use of GLP-1RA, which has substantial HbA1c-reduction properties [51]. In addition to their significance as glucose-lowering agents, GLP-1RAs present encouraging characteristics, including anti-inflammatory and anti-obesity properties, protective effects on the lungs, and a positive influence on the composition of the gut microbiome [52].

Nevertheless, GLP-1RAs are associated with common adverse gastrointestinal effects, which affect more than a third of patients [51].

The recently published STEP 1 (Semaglutide Treatment Effect in People with Obesity) [53], STEP 2 [54], STEP 3 [55], and STEP 4 [56], trials demonstrate the tremendous impact of weekly GLP-1RA treatment with semaglutide on weight loss and cardiovascular risk factors. The results of the trials led to the conclusion that 2.4 mg semaglutide once a week provides impressive weight loss and improvement in cardiovascular risk factors. It is approved for chronic weight management in adults with Body Mass Index (BMI) ≥ 30 kg/m^2^ or BMI ≥ 25 kg/m^2^ with concomitant conditions such as T2D, hypertension, or hyperlipidemia [57]. Additional metabolic benefits included improvements in liver fat content and reduced visceral and subcutaneous abdominal adipose-tissue volumes [21].

GLP-1RAs have additional benefits on blood pressure, with significant decreases in systolic blood pressure among hypertensive patients. Diastolic blood pressure is affected less. However, a significant increase in heart rate of 2–4 beats per minute has also been observed [58]. Furthermore, a meta-analysis published in November 2022 indicates an increased risk of both all thyroid cancer and medullary thyroid cancer associated with the use of GLP-1 RA, particularly after 1–3 years of treatment [59].

Cardiovascular outcome trials (CVOT) showed that treatment with GLP-1RA is associated with significant cardiovascular benefits [60].

In August 2023, Novo Nordisk disclosed the headline findings of the SELECT cardiovascular outcomes study. This double-blind trial investigated the use of subcutaneous, weekly 2.4 mg semaglutide in addition to standard care for the prevention of MACE over a period of up to five years, compared to a placebo. The study involved 17,604 adults ages 45 years or older who were diagnosed with overweight or obesity and established cardiovascular disease but who had no previous history of diabetes. The trial successfully met its primary objective by showing a statistically significant 20% reduction in MACE among individuals treated with 2.4 mg semaglutide in comparison to those who received the placebo. Real-world studies investigating cardio-renal outcomes of GLP-1RA suggest that initiation of GLP-1RA was associated with benefits for composite cardiovascular outcomes, MACE, all-cause mortality, myocardial infarction, stroke, cardiovascular death, and peripheral artery disease [60]. A study that examined the effects of the use of GLP-1 RA on stroke found that the use of semaglutide reduced the incidence of stroke compared to placebo among people with type 2 diabetes at high cardiovascular risk. This outcome was driven mainly by the prevention of small-blood-vessel blockage [61]. It has been shown that GLP 1 reduces proteinuria, but its effect on kidney function is still questionable. Some studies showed that GLP 1 increases cystatin C43, which raises concerns about possible negative renal outcomes [62]. Recently, the Icelandic Medicines agency reported about 150 cases of suicidal thoughts and self-injury among people using liraglutide and semaglutide medicines [63]. As a result, the EMA’s Safety Committee stated that it will review data on the risk of suicidal thoughts and self-harm as a potential adverse effect of GLP-1 RA use [64]. Clinical trials with GLP-1 RA such as SCALE and STEP usually exclude patients with major depressive disorders and previous suicidal attempts; thus, data about the increased risk of self-harm is limited. As of the writing of this review, we did not find compelling evidence in the literature of an increased risk of suicide and self-harm related to GLP-1 RA use [53,65]. Nonetheless, the EMA review is ongoing, and further recommendations cannot be made on this matter.

Oral GLP-1RA were first introduced in 2019 with oral semaglutide (Rybelsus). When compared with 1 mg subcutaneous semaglutide, there was no significant difference in weight loss and HbA1c at 20 mg and 40 mg doses, with standard escalation. The 2.5 mg, 5 mg and 10 mg groups were all inferior to 1 mg subcutaneous semaglutide in decreasing HbA1c (−0.4%, −0.9%, −1.2% and −1.9%, respectively). The 5 mg and 10 mg doses resulted in body weight decrease of −1.5 kg and −3.6 kg, respectively. Most adverse events are mild, with the most common being gastrointestinal disorders; these results are similar to those seen with the subcutaneous formulation [66].

In 2019, the PIONEER 6 trial was published as well. PIONEER 6 was a randomized, double-blind, placebo-controlled trial involving diabetic patients at high cardiovascular risk. A total of 3183 patients were randomly assigned to receive oral semaglutide or placebo. Major adverse cardiovascular events occurred in 3.8% of patients in the oral semaglutide group and 4.8% of patients in the placebo group (HR 0.79; 95% CI, 0.57 to 1.11; *p* < 0.001 for noninferiority). Gastrointestinal adverse events leading to discontinuation of oral semaglutide or placebo were more common with oral semaglutide [67].

In a randomized, placebo-controlled trial published in September 2023 that involved patients with heart failure with preserved ejection fraction and obesity, the administration of a weekly 2.4 mg dose of semaglutide resulted in more significant reductions in heart failure-related symptoms, physical limitations, and greater weight loss compared to the placebo over a 52-week period. Moreover, semaglutide also led to an increased 6-min walk distance, yielded more favorable outcomes in the evaluation of the hierarchical composite endpoint, and lowered C-reactive protein levels to a greater extent than the placebo did. Notably, the semaglutide treatment group experienced fewer serious adverse events when compared to the placebo group, and both groups had a similar frequency of discontinuation due to serious adverse events [3].

In a phase II, randomized, double-blind trial published in September 2023, a newly developed oral GLP-1RA named orforglipron showed promising results. The study included adults who either were diagnosed with obesity or who were overweight with at least one coexisting condition related to weight and who did not have diabetes. A total of 272 participants were randomly assigned. At the beginning of the trial, the average BMI was 37.9. By week 36, the average change ranged from −9.4% to −14.7% for those taking orforglipron, in contrast to a change of −2.3% observed with the placebo. The most frequently reported adverse events were gastrointestinal, which led to the discontinuation of orforglipron in 10% to 17% of participants across different dosage groups [68].

In recent years, promising novel treatment options in various research phases have joined the incretin mimetics drug class, including dual GLP-1/GIP agonists and GLP-1/glucagon agonists. GIP is a four-amino acid peptide produced by the K cells of the duodenum and jejunum. Like GLP-1, GIP is secreted rapidly after ingestion and stimulates insulin production in a glucose-dependent manner. GIP receptors are almost exclusively found in pancreatic β-cells, whereas GLP-1 receptors are found also in the α-cells of the endocrine pancreas, heart, central and peripheral nervous systems, stomach, and adipose tissue.

Tirzepatide, a dual GLP-1/GIP agonist administered once weekly, was approved in 2022 by the FDA for patients with T2D [2].

The SURPASS-2 trial, published in 2021, was a phase III, open-label trial wherein 1879 patients were randomly assigned to receive tirzepatide at 5 mg, 10 mg, or 15 mg doses or semaglutide at a dose of 1 mg. All three tirzepatide doses were superior to semaglutide regarding the mean decrease in HbA1c of - 0.15 percentage points (95% CI −0.28–−0.03; *p* = 0.02 for 5 mg), −0.39 percentage points (95% CI, −0.51–−0.26; *p* < 0.001 for 10 mg), and −0.45 percentage points (95% CI, −0.57–−0.32; *p* < 0.001 for 15 mg). Furthermore, the patients in the tirzepatide groups lost more weight than did those in the semaglutide group (least-squares mean estimated treatment difference, −1.9 kg, −3.6 kg, and −5.5 kg, respectively; *p* < 0.001 for all comparisons). Although the percentage of patients reporting any adverse event was similar across the groups (with GI events the most common), there was a significant increase in serious adverse events in the tirzepatide groups (5.3–7%) compared to the semaglutide group (2.8%) [69].

In 2022, the SURMOUNT-1 trial, a phase III, double-blind, randomized controlled trial was published. It included 2539 non-diabetic patients who were randomized to receive once-weekly, subcutaneous tirzepatide in 5 mg, 10 mg, or 15 mg dosages or placebo for 72 weeks, including a 20-week dose-escalation period. Tirzepatide was associated with a substantial reduction in body weight, with a mean percent decrease in weight at week 72 of −15.0% (95% CI, −15.9–−14.2) with 5 mg, −19.5% (95% CI, −20.4–−18.5) with 10 mg, and −20.9% (95% CI, −21.8–−19.9) with 15 mg. The mean percent decrease was −3.1% (95% CI, −4.3–−1.9) with placebo (*p* < 0.001 for all comparisons with placebo) [2].

In 2023, The “SURMOUNT-2” was published. This double-blind, randomized, multicenter, placebo-controlled, phase III trial assessed the efficacy and safety of tirzepatide versus placebo for weight management in people living with obesity and type 2 diabetes. This aim was in contrast to that of the “SURMOUNT-1” trial, wherein type 2 diabetes was a key exclusion criterion [70]. The co-primary endpoints were bodyweight reduction of 5% or more and percent change in body weight from baseline. A total of 938 participants were randomly assigned and received at least one 10 mg dose of tirzepatide (*n* = 312), 15 mg dose of tirzepatide (*n* = 311), or placebo (*n* = 315). Least-squares mean changes in body weight at week 72 with 10 mg and 15 mg tirzepatide were −12.8% (SE 0.6) and −14.7% (0.5), and −3.2% (0.5), respectively, resulting in an estimated treatment differences versus placebo of −9.6% (95% CI −11.1–−8.1) with 10 mg tirzepatide and −11.6% (−13.0–−10.1) with 15 mg tirzepatide (all *p* < 0.0001). From a baseline HbA1c of 8.02% (64.1 mmol/mol), HbA1c improved by −2.1% (SE 0.06) with 10 mg tirzepatide, −2.1% (SE 0.07) with 15 mg tirzepatide, and −0.5% (SE 0.07) with placebo (*p* < 0.0001 for all comparisons vs. placebo). At week 72, the mean HbA1c values were 6.0%, 5.9%, and 7.5%, with 10 mg tirzepatide, 15 mg tirzepatide, and placebo, respectively. The proportion of participants in each group reaching HbA1c levels of less than 7.0%, 6.5% or less, or less than 5.7% was significantly higher in the 10 mg and 15 mg tirzepatide groups compared with placebo (87% [*n* = 271] and 84% [*n* = 262] vs. 36% [*n* = 114], 80% [*n* = 249] and 79% [*n* = 247] vs. 20% [*n* = 63], and 46% [*n* = 144] and 49% [*n* = 151] vs. 4% [*n* = 12], respectively [70,71].

Retatrutide, another novel incretin-mimetic drug, is an agonist of the GLP-1, GIP and glucagon receptors. The Retatrutide Phase 2 trial, published in August 2023, was a phase II, double-blind, randomized, placebo-controlled trial involving 338 non-diabetic adults who had a BMI of 30 or higher, or a BMI of 27 to less than 30 and at least one weight-related condition. Participants were randomized to administration of subcutaneous retatrutide at varying doses (1 mg, 4 mg [initial dose: 2 mg], 4 mg [initial dose: 4 mg], 8 mg [initial dose: 2 mg], 8 mg [initial dose: 4 mg], or 12 mg [initial dose: 2 mg]), or a placebo, weekly, over 48 weeks. At 48 weeks, the least-squares mean percentage weight changes in the retatrutide groups were as follows: −8.7% in the 1 mg group, −17.1% in the combined 4 mg group, −22.8% in the combined 8 mg group, and −24.2% in the 12 mg group, compared to −2.1% in the placebo group. Weight reductions of 5% or more, 10% or more, and 15% or more were observed in 92%, 75%, and 60% of participants receiving 4 mg of retatrutide; 100%, 91%, and 75% for those on 8 mg; 100%, 93%, and 83% for those on 12 mg; and 27%, 9%, and 2% for those receiving placebo. The predominant adverse events in the retatrutide groups were gastrointestinal. These adverse events were dosage-dependent, were mainly of mild to moderate intensity, and were somewhat alleviated with a lower initial dosage (2 mg versus 4 mg) [72].

## 9. Sodium-Glucose Cotransporter Inhibitors (SGLT-2i)

SGLT-2 inhibitors are a class of medications primarily used for managing type 2 diabetes. They provide insulin-independent glucose lowering by blocking glucose reabsorption in the proximal renal tubules, consequently lowering blood glucose levels. One of the significant benefits of SGLT-2i is their association with weight loss due to the elimination of excess glucose. Moreover, these drugs are associated with consistent reductions in blood pressure. They have a relatively low risk of causing hypoglycemia compared to other diabetes medications, making them a safer option, especially in combination with other anti-diabetic drugs. Beyond their glucose-lowering effects, SGLT-2i exhibit notable cardiovascular benefits, as demonstrated by a reduced risk of heart failure and cardiovascular events. Furthermore, they have shown promise in slowing the progression of kidney disease in diabetic patients. These conclusions are drawn from reliable clinical trials and studies, positioning SGLT-2i as a valuable addition to the management of type 2 diabetes, with potential multifaceted health benefits [1,10,21,73,74,75].

Cardiorenal outcome trials have shown the effectiveness of these drugs in lowering the risk of MACE, HHF, all-cause mortality, and renal deterioration [1,5,21,75].

Furthermore, SGLT-2i improve the outcomes of patients with HF or chronic kidney disease (CKD) regardless of whether they have T2DM or not [1,5,75].

SGLT-2i may reduce the risk of cardiovascular death or HHF for both diabetic and non-diabetic patients with HF with reduced ejection fraction (HFrEF) [5,75].

Additionally, in patients with HF with preserved ejection fraction (HFpEF), SGLT-2i reduce the combined risk of cardiovascular death or HHF and improve symptoms related to HF and physical limitations [76,77,78].

Empagliflozin and dapagliflozin are also associated with an improvement in left-ventricle structure and function in patients with HF, but more information is required to further clarify these effects [79,80].

In patients with CKD, SGLT-2i use is associated with a reduction in the decline of glomerular filtration rate (GFR), proteinuria, end-stage renal disease, and the risk of death from renal or cardiovascular causes [1].

In a 2020 study published in the New England Journal of Medicine, 4304 participants with an estimated GFR ranging from 25 to 75 mL per minute per 1.73 m^2^ of body surface area and a urinary albumin-creatinine ratio (with albumin measured in milligrams and creatinine measured in grams) in the range of 200 to 5000 were administered either 10 mg dapagliflozin daily or a placebo. For patients with chronic kidney disease, irrespective of the presence or absence of diabetes, the study revealed that the risk of the composite outcome, including a sustained decline in estimated GFR of at least 50%, end-stage kidney disease, or death from renal failure, was significantly lower in the group receiving dapagliflozin compared to that in the group receiving the placebo (95% CI, 0.45–0.68; *p* < 0.001). The HR for the composite outcome of death from cardiovascular causes or HHF was 0.71 (95% CI, 0.55–0.92; *p* = 0.009) [1].

A recent meta-analysis of randomized controlled trials compared the efficacy and safety of various drug treatments for T2DM and found that compared to other drugs, SGLT-2i increased the risk of diabetic ketoacidosis (DKA) (OR 2.07, 95% CI 1.44–2.98), genital infection (OR 3.30, 95% CI 2.88–3.78) and probably amputation (OR 1.27, 95% CI 1.01–1.61) [81].

The DKA associated with SGLT-2i may manifest as euglycemic DKA in some patients with T2DM, specifically in those with a glucose level of less than 200 mg/dL. This potential adverse effect requires vigilance from healthcare providers, as euglycemic DKA can be life-threatening and is easily missed [82].

Canagloflizin may increase the risk for amputation and fractures [83,84,85]. It has been suggested that other SGLT-2i may also increase the risk of bone fractures, but this finding was not confirmed in subsequent meta-analyses, although additional data are required to establish a firm conclusion [86,87].

In a meta-analysis conducted for patients with NAFLD, SGLT-2i and GLP-1 RA were studied along with thiazolidinediones. These drugs were found to improve liver enzymes, BMI, blood lipid levels, blood glucose and insulin resistance in patients with NAFLD [74].

Considering the many advantages mentioned herein, SGLT-2i are recommended as a first-line therapy for T2DM with CVD or renal disease by the American Diabetes Association (ADA) and the European Association for the Study of Diabetes (EASD) [21].

## 10. Is the Combination of SGLT-2i and GLP-1 RA Synergistic?

Robust evidence suggests that SGLT-2i and GLP-1RAs are associated with low risk of hypoglycemia; promote weight loss; and exert a positive impact on vascular, cardiac, and renal endpoints. Several post hoc analyses of cardiovascular outcomes studies involving diabetic patients already treated with GLP1-RA have suggested that the combination of SGLT-2i and GLP-1 exhibits a synergistic effect in protecting the heart and kidney [7,21,40,60].

## 11. Pharmacotherapy in Pregnancy

Among normo-glycemic women, early pregnancy is characterized by a notable increase in insulin sensitivity. However, the opposite occurs as pregnancy progresses to the second and third trimesters, when insulin sensitivity decreases substantially. This change results in a decrease in glucose uptake by adipose and muscle tissue, as well as decreased insulin-mediated lipolysis and beta-oxidation. In tandem, these adaptations help to meet the surging energy demands of the developing fetus [88].

In women with pre-existing diabetes, these changes in glucose metabolism are more pronounced and may lead to a hyperglycemic state, which makes this population more susceptible to pregnancy complications [88], including congenital malformations, preeclampsia, and preterm delivery. In addition, about half of the women with pre-existing diabetes deliver infants who are large for gestational age, which elevates the infants’ risk of birth-related traumas and the risk of developing metabolic syndrome, cardiovascular disease, and type 2 diabetes mellitus later in life [89]. Therefore, glycemic control is of paramount importance in pregnant women with pre-existing diabetes.

The glucose targets for pregnant women with pre-existing diabetes or gestational diabetes mellitus (GDM) recommended by the ADA, as well as the American College of Obstetricians and Gynecologists (ACOG), are fasting glucose, ≤95 mg/dL; glucose 1 h after eating, ≤140 mg/dL; and glucose 2 h after eating, ≤120 mg/dL [90,91]. Although HbA1c is a useful measure of glycemic control in a non-pregnant population, in pregnancy, it seems to less accurately reflect temporary glucose fluctuations due to an increased turnover of red blood cells [89]. In addition, elevations in fasting glucose, in contrast to postprandial elevations, are more predictive of fetal macrosomia. This insight can prove valuable when adjusting insulin regimens [92].

Managing T2DM with hypoglycemic agents, including insulin, during pregnancy, along with proper diet and lifestyle modifications to ensure glycemic control, are paramount for the prevention of pregnancy complications [89].

Pharmacologic treatment of T2DM in pregnancy is based primarily on insulin. Nevertheless, non-insulin hypoglycemic agents are available and include metformin and glyburide. Other classes of diabetic medications are not recommended in pregnancy due to the limited safety and efficacy data available for this population [93].

Both ACOG and the ADA recommend that insulin therapy should be the cornerstone of pharmacotherapy and that the use of other agents should be limited or ceased entirely during pregnancy [88,89].

Metformin is a biguanide that exerts its antihyperglycemic effects by decreasing hepatic gluconeogenesis, decreasing glucose absorption in the intestines, and increasing glucose uptake in the periphery [89]. It is inexpensive and effective in reducing hyperglycemia and is the first-line therapy for many patients with T2DM [94]. In pregnant women, metformin seems to improve maternal metabolic outcomes. The Metformin in Women with Type 2 Diabetes in Pregnancy study (MiTy), a multicenter, randomized, controlled trial, compared 2000 mg daily metformin with placebo. Women in the metformin group achieved better glycemic control compared to those given the placebo, as manifested by reduced HbA1c levels (5.9% [SD 0.78] vs. 6.1% [SD 0.94], *p* = 0.015), lower insulin requirements, less weight gain, and fewer cesarean sections [95]. Nevertheless, metformin, in contrast to insulin, can cross the placenta and potentially affect the growing fetus, which raised concerns regarding its use in pregnancy. A meta-analysis that looked at the use of metformin in the first trimester of pregnancy did not find a significant increase in major congenital malformations [96]. In addition, metformin may reduce birthweight, thereby decreasing the rates at which infants are born large for gestational age and/or with macrosomia and increase the rate of at which infants are born small for gestational age, which is also associated with adverse outcomes [95,97]. In contrast to its effects on birthweight, exposure to metformin in utero is associated with a higher BMI later in life [96,98]. In contrast to these findings, a 24-month follow-up on the children born to women in the MiTy trial concluded that there was no difference in BMI and overall anthropometrics between the children who were exposed to metformin or placebo [99]. In summary, although metformin seems to improve maternal metabolic outcomes in pregnancy, its effects on the fetus are still not clear and additional long-term data are needed to establish a firm recommendation.

Glyburide, also known as glibenclamide, is a long-acting sulfonylurea that exhibits antihyperglycemic effects by stimulating the release of insulin from pancreatic beta cells and increasing peripheral insulin sensitivity. Like metformin, glyburide can cross the placenta and potentially affect a growing fetus [43]. Multiple meta-analyses sought to compare the safety and efficacy of glyburide with metformin therapy in pregnancy, as these are the only oral diabetic agents currently available for this population and are commonly prescribed in combination with insulin [100,101,102,103]. Glyburide is about equal to metformin in its contribution to glycemic control in pregnant women with diabetes. A meta-analysis that compared glyburide and metformin in patients with GDM found no difference in fasting blood glucose (*p* = 0.821), postprandial blood glucose (*p* = 0.217) and birth weight (*p* = 0.194) between the interventions. However, glyburide is associated with more adverse effects, including maternal weight gain, macrosomia, neonatal hypoglycemia, and neonatal disease [100]. When safety and efficacy are evaluated, glyburide seems to be inferior to metformin in the treatment of diabetes in pregnancy because it has more adverse effects and only comparable efficacy, with a lack of long-term data on its effects on the fetus.

Insulin and insulin analogs are the mainstay of treatment for pregnant diabetic women and are universally prescribed for this population [90,93]. Insulin does not cross the placenta [90] and thus is deemed safer for the fetus than oral antihyperglycemic agents; however, the evidence that oral agents are not safe is not convincingly found in the literature due to limited data. Insulin requirements change with gestational age. Therefore, frequent monitoring and adjustments are important to improve efficacy and safety of treatment [90]. In practice, a combination of long-acting and short-acting insulin is used to establish glycemic control. Long-acting insulin creates a sufficient baseline level of insulin, while short-acting insulin accommodates postprandial increases in glucose [93]. Long-acting agents include detemir, glargine, and degludec, and short-acting agents include lispro, aspart, and glulisine.

Insulin analogs are considered safe and efficient for treating diabetes in pregnancy. In a meta-analysis of randomized, controlled trials that compared detemir to neutral protamine Hagedorn (NPH) insulin, detemir was associated with fewer maternal hypoglycemic events and decreased risk of prematurity. Another multicenter, prospective cohort compared detemir to other basal insulins including NPH in pregnant women with pre-existing diabetes and found no difference in efficacy and safety [104,105]. The recently published EXPECT trial found degludec to be non-inferior to detemir, while data on glargine is mostly observational and requires further investigation [106]. Lispro and aspart are at least as safe as NPH. They are better than NPH in postprandial glucose regulation and may create a synergistic effect with long-acting insulin to provide good glycemic control [107]. Currently, data regarding the use of glulisine in pregnancy is lacking compared to data regarding lispro and aspart. It should therefore be prescribed with more caution. Multiple daily injections (MDI) are the standard modality of insulin administration in diabetic pregnant women, but continuous subcutaneous insulin infusion (CSII) with an insulin pump is another available option. Current evidence suggests no advantage to the CSII modality [90,108].

As current evidence stands, insulin supplied in a basal-bolus manner is the first-line pharmacologic treatment for diabetic pregnant women. While metformin may confer some additional benefit, additional long-term data about the effects on offspring are needed to establish a clear recommendation.

## 12. Breakthrough Research and Future Drugs

In a study examining the molecular basis of insulin resistance, researchers identified a small molecule, Co-Insulin, which binds specifically to insulin rather than to the insulin receptor (IR). This 222 Da molecule binds to a specific site on the A-chain of insulin, initiating interaction between the A and B chains and activating the insulin receptor. This activation triggers a series of events: insulin receptor autophosphorylation, phosphorylation of IRS proteins, and insulin action with ‘Immediate’ and ‘Delayed’ responses. The ‘Immediate’ response involves the known hypoglycemic effect mediated by Glut-4 and influenced by a phosphorylated IRS molecule. Conversely, the ‘Delayed’ response involves the nuclear entry of specific fragments of the phosphorylated insulin receptor, which is potentially facilitated by another phosphorylated IRS. Importantly, Co-Insulin functions solely by initiating insulin signaling events. This pioneering study raises the question of whether similar small molecules play roles in mediating the actions of other peptide hormones in the mammalian system. This research may lead to the development of new diabetes drugs [109].

## 13. Innovation in the Therapeutic Approach

The importance of normalized blood glucose levels was raised in the DPP-1 study of patients with pre-diabetes [110]. The study evaluated whether healthy lifestyle or metformin led to normalized blood glucose levels in some of the study participants. Even temporary normalization of blood glucose levels was shown to delay the onset of diabetes by several years [110].

Recent studies have highlighted a clear correlation between reducing HbA1c to less than 6.5% and a significant decrease in microvascular and macrovascular complications [66]. When we combine these findings, emphasizing the importance of normalizing blood glucose levels to prevent both the development of diabetes and associated risk factors, our current goals should be to cure diabetes and to aim to lower HbA1c below 5.7%, even if the effect lasts only for several years. This approach could significantly impact outcomes for patients with T2DM, especially considering that most individuals develop the disease after age 50 [66].

The new drugs, which not only lower glucose levels but also reduce cardiovascular risk factors, promote weight loss, and reduce abdominal fat, offer the potential to cure diabetes in a sizable proportion of patients, provided that we diagnose the condition as early as possible. Therefore, the treatment approach should be substantially different from traditional methods. Patients with either pre-diabetes or diabetes should be diagnosed as early as possible. Diabetic patients should be treated as early as possible with healthy-lifestyle interventions and combination therapy including at least one of these new drugs.

Recent studies and meta-analyses suggest the potential benefits of primary combined therapy for glycemic management in diabetes compared to monotherapy with metformin across a wide range of baseline HbA1c levels. Combined treatment can increase the number of patients achieving their glycemic goals. The newer glucose-lowering drugs reduce the risk of hypoglycemia and help reduce patient weight. Furthermore, improving our understanding of the complex pathophysiology of T2DM and the availability of therapies that target specific mechanisms contributing to hyperglycemia should lead to increased adoption of combined treatment, which is pathophysiologically correct. This approach to achieving better glycemic control (both in the short term and in the long term) helps balance the side effects associated with monotherapy and results in better glycemic control. In addition, better balance is maintained over both the short term and the long term [7]. Early diabetes control has a wide-ranging impact on the prevention of disease complications throughout life [62].

The most effective way to manage diabetes is to treat it with a combination therapy from the time of initial diagnosis.

Recovery from diabetes includes two important aspects: weight loss and normalization or near-normalization of blood glucose levels.

The recommendation for optimal glucose balance includes commencing treatment with dual therapy. The question is whether to use metformin together with GLP-1 RA or SGLT-2i, or to initiate combined treatment with GLP-1 RA and SGLT-2i.

If it is not contraindicated, we believe that all diabetic patients with or without cardiovascular or renal disease will benefit from using metformin as part of the combination therapy, together with SGLT-2i or GLP-1 RA. We recommend considering combined treatment with metformin, GLP-1 RA, and SGLT-2i in patients who do not achieve the target of HbA1c or weight loss or who are at increased risk of cardio-renal events, as this has been proven to reduce complications and mortality among patients with these conditions.

There are certain indications for initial therapy with a three-drug combination, including prior failure to achieve success with a two-drug combination, patients with active cardiovascular disease or heart failure, and patients with proteinuria and deterioration in kidney function.

To help medical staff choose the ideal drug combination for each patient, the National Diabetes Council in Israel developed a calculator that recommends the most appropriate combination therapy based on the patient’s BMI, HbA1c levels, and risk for ischemic heart disease, stroke, renal function, and heart failure.

Along with the most appropriate drug treatment, a healthy lifestyle is extremely important. The keys to a healthy lifestyle are exercise, maintaining an appropriate diet, and 7–8 h of sleep a night [111].

Five distinct types of diabetes were recently characterized [112]. Some include issues related to deficient beta-cell mass and function. In these cases, the addition of insulin, sulfonylurea, glinide, and in some instances, TZD or DPP4-i, can maintain the target blood glucose levels for a significant portion of these patients. Consequently, there is a strong case for combining these drugs in the treatment of diabetic patients. This recommendation may be especially important in the coming years, particularly given the high cost of the new drugs, which restricts their use in many patients. Short-term treatment with insulin is recommended for patients with uncontrolled diabetes resulting in clinical symptoms and/or HbA1c greater than 9% (Figure 1).

Hence, our approach to treating recently diagnosed diabetes is as follows:Early diagnosis of abnormal glucose metabolism.Early treatment with combination therapy.Giving preference to medications that simultaneously lower weight and blood glucose levels in those who are overweight or obese, do not cause hypoglycemia, and protect against possible damage to target organs like the heart, the macro- and microvascular systems, the kidneys, the liver, and the brain.Aiming to normalize blood glucose levels using the new drugs and adding the old ones for patients who do not achieve the target HbA1c.

## Figures and Tables

**Figure 1 ijms-24-17147-f001:**
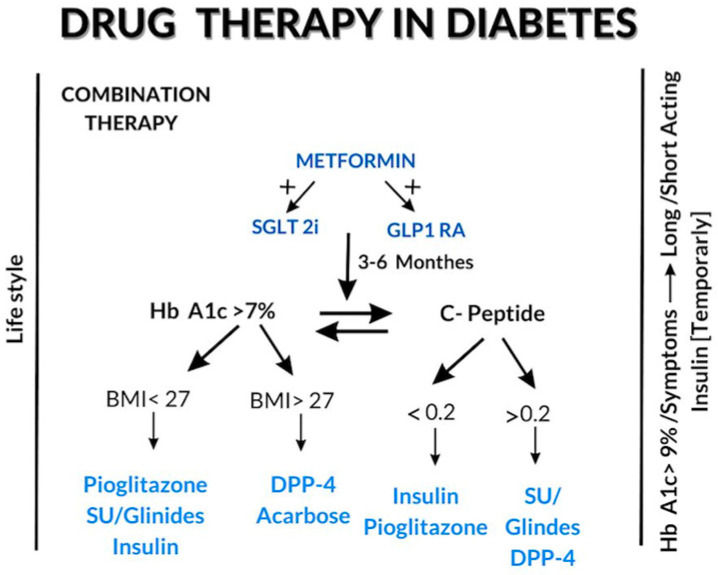
Drug-Treatment Algorithm for Diabetes. The algorithm outlines our recommendations for diabetes treatment, incorporating the initiation of combined drug therapy and adjusting medications based on the patient’s response.

**Table 1 ijms-24-17147-t001:** Anti-Diabetic Drugs. The table provides an overview of oral and injectable non-insulin medications used to improve blood glucose levels. It includes the names of medication groups and specific drugs within each group, as well as information on their effects, side effects, and safety. (‘↑’Aggravation/increase, ‘↓’ improvement/decrease, ‘?’ Not confirmed).

Drug Group	Specific Drug	Effects	Adverse Effects	Safety
Biguanides	Metformin	HbA1c ↓ Body weight ↓→ Cancer ↓? Cardiovascular↓?	Gastrointestinal disorders ↑ Reversible vitamin B12 deficiency↑ Lactic Acidosis ↑	None
Glinides	Repaglinide Nateglinide	HbA1c ↓ Body weight ↑	Hypoglycemia ↑ Headache ↑ Upper respiratory tract infection ↑	None
Alpha-Glucosidase inhibitors	Acarbose	HbA1c ↓ Body weight ↓ →	Gastrointestinal disorders ↑ Serum transaminases (AST, ALT) ↑	None
SGLT2-I	Empagliflozin Dapagliflozin Canagliflozin	HbA1c ↓ Body weight ↓ BP ↓ MACE ↓ Hospitalization for HF ↓ Progression of renal disease ↓	Diabetic ketoacidosis ↑ Genital infection ↑ Urinary tract infection ↑ Hypovolemia ↑ Acute kidney injury ↑ (related to hypovolemia) Canagliflozin: Amputation ↑ Bone fracture ↑	None
Thiazolidineidiones	Pioglitazone	HbA1c ↓ BP ↓ NAFLD ↓ MACE ↓	Body weight ↑ Peripheral edema ↑ Anemia ↑ Hospitalization for HF ↑ Bone fracture in women ↑	Cancer?
DPP-4	Sitagliotin Saxagliptin Alogliptin	HbA1c ↓	Saxagliptin: Hospitalization for HF?	
Sulfonylureas	Glimepiride Gliclazide Glibenclamide Glipizide	HbA1c ↓	Body weight ↑ Hypoglycemia ↑ Lack of durable effect	Glibenclamide Glipizide: Cardio-vascular events?
GLP-1 RA Combination therapy	Liraglutide Dulaglutide Semaglutide Orforglipron Tirzepatide Retatrutide	HbA1c ↓↓ Body weight ↓ Systolic BP ↓ MACE ↓ HF ? Quality of life for patients with HF (KCCQ-CSS) ↑ NAFLD ↓ →	Gastrointestinal disorders ↑ Semaglutide: Macular edema	Pancreatitis Bile stones Thyroid carcinoma

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
