# Peer review of "Drug Therapies for Diabetes"

_ijms, 2023, doi:10.3390/ijms242417147_

Round 1
Reviewer 1 Report
Comments and Suggestions for Authors
Dear Author(s),
Thank you for your interesting manuscript, that is not novel, but it is adequately summarized. However, to make it a contribution to the scientific community, I suggest the following modifications:
-Organize a paragraph for each medication group: Discuss on clinical trials that led to registration of the agents - present their primary otucomes such as effectiveness regarding HbA1c reduction, and also discuss on related CVOT outcomes. For each drug comment on effectiveness in terms of HbA1c reduction as a surogate marker (bearing in mind firstly RCTs, then RWDs and finally pooled data from meta-analyses), then comment on potential cardiovascular (MACE), HF, and renal benefits, or even additional ones (e.g. NAFLD and pioglitazone). In addition, talk on potential considerations and safety data (even novel adverse events as a new signal on thyroid carcinoma from PRAC few weeks ago from PRAC that is under investigation; but also on all important ones - contraindications relative and absolute and adverse events and risk of hypoglycaemia event with citing the relevant pharmacoepidemiology and pharmacovigilance data for all domains). Then talk on cost-effectiveness perspective and cost of medication groups.
-You should also make a paragraph for terzipatide (GLP-1RA/GIP agent) and also for insulin regimens, since you are talking about all the medicines for T2D management.
-A brief intro into the topic would be beneficial as well as mentioning the aim at the end of the introduction.
-Do you have a consent for publication of the figures that are taken from the joint ADA/EASD gudielines?
-After this modificiations and in general, please discuss what is novel and why is your paper a substantial scientific contribution that should be published?
-Consider commenting on SGLT-2 cardiorenal benefits (e.g. https://doi.org/10.3390/diabetology4030022) in more detail, and also consider mentioning other (secondary) benefits of GLP-1RA therapy (e.g. 10.1111/cob.12439).
-What are the future potentials in the management of T2D to the best of your knowledge? Consider adding a comment (e.g. Diabetology | Free Full-Text | Current Obstacles (With Solutions) in Type 2 Diabetes Management, Alongside Future Directions (mdpi.com)).
Best regards, Reviewer
Author Response
Reviewer 1
Thank you for your careful review of our manuscript. We welcome your comments and edits below and have incorporated them carefully to improve the manuscript.
An itemized point-by-point response to your comments is presented below.
Thank you for your interesting manuscript, that is not novel, but it is adequately summarized. However, to make it a contribution to the scientific community, I suggest the following modifications:
-Organize a paragraph for each medication group: Discuss on clinical trials that led to registration of the agents - present their primary otucomes such as effectiveness regarding HbA1c reduction, and also discuss on related CVOT outcomes. For each drug comment on effectiveness in terms of HbA1c reduction as a surogate marker (bearing in mind firstly RCTs, then RWDs and finally pooled data from meta-analyses), then comment on potential cardiovascular (MACE), HF, and renal benefits, or even additional ones (e.g. NAFLD and pioglitazone). In addition, talk on potential considerations and safety data (even novel adverse events as a new signal on thyroid carcinoma from PRAC few weeks ago from PRAC that is under investigation; but also on all important ones - contraindications relative and absolute and adverse events and risk of hypoglycaemia event with citing the relevant pharmacoepidemiology and pharmacovigilance data for all domains). Then talk on cost-effectiveness perspective and cost of medication groups.
We incorporated a summary table for each drug, delineating its efficacy in reducing HbA1c, MACE, cardiovascular benefits, renal benefits, HF, and cancer among other aspects. Moreover, we detailed the primary advantages and disadvantages of each drug within the text. We also explored the associated risks, safety data, and cost-effectiveness. We added descreption of key cardiovascular and renal outcomes trials.
(Table 1, Lines: 29-32, 78-82, 107- 118, 175-178, 285-287, 421-431, 447-457)
-You should also make a paragraph for terzipatide (GLP-1RA/GIP agent) and also for insulin regimens, since you are talking about all the medicines for T2D management.
Thank you for your suggestion. We agree that delving into teriparatide was essential. Consequently, we included a paragraph on this specific subject. As the focus the review on oral non insulin injuctable drugs for diabetes treatment, we deliberately excluded information pertaining to insulin regimens.
(Lines 358-399)
-A brief intro into the topic would be beneficial as well as mentioning the aim at the end of the introduction.
Thank you for highlighting this point. In accordance with your insightful comment, we included an introduction paragraph that elaborates on the topic and outlines the purpose of the review.
(Lines 28 - 51)
-Do you have a consent for publication of the figures that are taken from the joint ADA/EASD gudielines?
After revising the paper, we replaced the existing guideline figures and added a new proposal.
(Figure 1)
-After this modificiations and in general, please discuss what is novel and why is your paper a substantial scientific contribution that should be published?
We added a paragraph that highlights the paper's significance, illustrating its novelty, and providing treatment suggestions
(Lines: 601-588)
-Consider commenting on SGLT-2 cardiorenal benefits (e.g. https://doi.org/10.3390/diabetology4030022) in more detail, and also consider mentioning other (secondary) benefits of GLP-1RA therapy (e.g. 10.1111/cob.12439).
We combined assessments of the cardiac benefits of SGLT2 inhibitors and expanded the literature on the additional and secondary benefits of GLP-1RA treatment.
(Lines: 266-270, 331-340, 380-399, 418-431, 433-435, 447-457)
-What are the future potentials in the management of T2D to the best of your knowledge? Consider adding a comment (e.g. Diabetology | Free Full-Text | Current Obstacles (With Solutions) in Type 2 Diabetes Management, Alongside Future Directions (mdpi.com)).
Thank you for the excellent comment and referral; it helped us substantially. We included a paragraph explaining the innovation of the paper, discussing the future potential in managing type 2 diabetes, presenting existing obstacles, and proposing a new treatment algorithm.
(Lines: 614 - 620, 636-677, Figure 1)

Reviewer 2 Report
Comments and Suggestions for Authors
Manuscript ID: ijms-2657080
Type of manuscript: Review
Title: Drug Therapies for Diabetes
The review article entitled, ‘Drug Therapies for Diabetes’ is a comprehensive one. This reviewer would advise some modifications amounting to a major revision. After such a revision it is recommended for acceptance.
The following major changes are suggested:
1. A total account of the history of each drug may be briefly given; particularly for metformin, which has a chequered history. Metformin has had a complex history with periods of controversy and withdrawal, followed by resurgence and widespread use due to its efficacy and safety. It remains a crucial medication for managing type 2 diabetes, and ongoing research is exploring its potential benefits in other health-related areas. The chequered history of metformin underscores the dynamic nature of medical science and the need for ongoing research. The review should reflect this. Unfortunately, the review article under review lacks the above rigor.
2. In the case of each drug, the current status of the molecular basis of drug action may be briefly attempted. For example, the case of metformin may be reconsidered in the light of a recent article, “Marc Foretz, Bruno Guigas & Benoit Violle (2023). Nature Reviews Endocrinology, Review article, https://doi.org/10.1038/s41574-023-00833-4”.
3. Wherever possible review may be done on the attempts to improve the drug quality, based on the receptor/target structure-based designing. This is optional for the authors.
4. If the authors are suggested to go through the article, “Thampan, R.V., Krishnaraj, K.U., Ali, H.S. et al. Insulin Signalling: Essential Role of a 222 Da Molecular Mediator, Co-Insulin (Co-Ins). Proc. Natl. Acad. Sci., India, Sect. B Biol. Sci. 90, 843–853 (2020). https://doi.org/10.1007/s40011-019-01157-y
Minor Changes
The text must be thoroughly edited and copy-edited. Some such errors are shown below:
a. In many a place ‘trial’ has become ‘trail’. For example; lines, 279 (two times,), L 291 (two times,), and more
b. Inconsistency of denoting measures and units: lines, 305, 306: 2 mg, 4 mg, and many others. Line 390: 95mg/dL; and
Comments on the Quality of English LanguageEnglish editing and copy editing (science) are needed. The authors themselves may be able to do so.
Author Response
Reviewer 2
The review article entitled, ‘Drug Therapies for Diabetes’ is a comprehensive one. This reviewer would advise some modifications amounting to a major revision. After such a revision it is recommended for acceptance.
Thank you for your positive feedback. We welcome your comments and edits below and have incorporated them carefully to improve the manuscript.
An itemized point-by-point response to your comments is presented below.
The following major changes are suggested:
- A total account of the history of each drug may be briefly given; particularly for metformin, which has a chequered history. Metformin has had a complex history with periods of controversy and withdrawal, followed by resurgence and widespread use due to its efficacy and safety. It remains a crucial medication for managing type 2 diabetes, and ongoing research is exploring its potential benefits in other health-related areas. The chequered history of metformin underscores the dynamic nature of medical science and the need for ongoing research. The review should reflect this. Unfortunately, the review article under review lacks the above rigor.
We agree with this suggestions, we expended the pharagraph on metformin and included a brief description before each drug, outlining its notable advantages, disadvantages, and current usage. We did not go into details because of the length of the review and the large amount of references.
(Lines: 66-77, 128-134, 168-173, 418-431)
- In the case of each drug, the current status of the molecular basis of drug action may be briefly attempted. For example, the case of metformin may be reconsidered in the light of a recent article, “Marc Foretz, Bruno Guigas & Benoit Violle (2023). Nature Reviews Endocrinology, Review article, https://doi.org/10.1038/s41574-023-00833-4”.
Given that this paper is primarily intended for clinicians, it is crucial that we highlight the clinical benefits and drawbacks of each drug while providing practical treatment recommendations. At the same time, we added a brief description of the molecular basis in each drug, and after reviewing the excellent article you shared, incorporated additional information about metformin, which significantly contributed to our understanding of the drug's complexity and its treatment.
(Lines: 66-77, 128-134, 168-173, 418-431)
- Wherever possible review may be done on the attempts to improve the drug quality, based on the receptor/target structure-based designing. This is optional for the authors.
Thank you for your suggestion; it is indeed intriguing. Simultaneously, our focus is on outlining the drug's pros and cons while offering comprehensive treatment guidelines, rather than delving into its molecular structure, specific drug intricacies, and similar technical details.
- If the authors are suggested to go through the article, “Thampan, R.V., Krishnaraj, K.U., Ali, H.S. et al.Insulin Signalling: Essential Role of a 222 Da Molecular Mediator, Co-Insulin (Co-Ins). Proc. Natl. Acad. Sci., India, Sect. B Biol. Sci. 90, 843–853 (2020). https://doi.org/10.1007/s40011-019-01157-y
After reading the insightful article you shared, we added a section in the paper titled 'Breakthrough Research and Future Drugs.' This part explores the themes highlighted in the referenced article and examines the potential for the development of new medications.
(Lines 587-600)
Minor Changes
The text must be thoroughly edited and copy-edited. Some such errors are shown below:
- In many a place ‘trial’ has become ‘trail’. For example; lines, 279 (two times,), L 291 (two times,), and more
Thank you very much for the attention, it has been revised.
- Inconsistency of denoting measures and units: lines, 305, 306: 2 mg, 4 mg, and many others. Line 390: 95mg/dL; and
Thank you for pointing these out. We corrected them.
Comments on the Quality of English Language
English editing and copy editing (science) are needed. The authors themselves may be able to do so.
Thank you for your comment, the paper has been reviewed by a medical editor.
Reviewer 3 Report
Comments and Suggestions for Authors
The draft is well-structured and comprehensive, offering a detailed overview of Type 2 diabetes treatments and their implications. However, here are some suggestions to enhance clarity, accuracy, and flow:
1. Line 38-39: Pay attention to sentence structure for clarity, especially in complex sentences. For example, in the sentence "Regarding dementia, a meta-analysis presented that cognitive impairment was significantly less prevalent...," consider rephrasing for smoother readability.
2. Line 34-36: The potential protective effects of metformin on the cardiovascular system, kidney function, and neurodegenerative conditions are intriguing and relevant. It would be beneficial to elaborate on the specific findings of the meta-analysis or studies that support these claims.
3. Line 43-50: The explanation of metformin's potential role in cancer treatment is insightful, particularly the mention of its ability to reduce circulating insulin levels and its direct inhibitory effects on cancer cells. Providing some specific examples or statistics on the success of metformin in cancer treatment would strengthen this section.
4. Line 61-63: The statement about a consensus on other first-line treatment approaches for certain patients is relevant. If possible, provide examples or criteria for when alternative treatments might be considered.
5. Line 175: "Acrabose (Precose)" should be corrected to "Acarbose (Precose)".
6. Line 175-177: There's an inconsistency with the dates. The FDA approved Acarbose in 1999, but the clinical trial was published in 1995. If Acarbose was only approved in 1999, how was a trial published in 1995? This needs clarification.
- Line 185: "incase" should be two words: "in case."
- Please fix the references in the manuscript uniformly
- Please reduce the similarities level from the manuscript.
The overall quality of English language and grammar in the draft is quite good. The sentences are well-structured, and the language used is clear and concise. However, there are a few minor grammatical issues and areas where language could be further improved for clarity.
Author Response
Reviewer 3
The draft is well-structured and comprehensive, offering a detailed overview of Type 2 diabetes treatments and their implications. However, here are some suggestions to enhance clarity, accuracy, and flow:
Thank you for your positive feedback. We welcome your great comments and suggestions and have incorporated them carefully to improve the manuscript.
- Line 38-39: Pay attention to sentence structure for clarity, especially in complex sentences. For example, in the sentence "Regarding dementia, a meta-analysis presented that cognitive impairment was significantly less prevalent...," consider rephrasing for smoother readability.
Thank you for your comment. It has been revised, and the text has been reviewed by a professional medical editor.
- Line 34-36: The potential protective effects of metformin on the cardiovascular system, kidney function, and neurodegenerative conditions are intriguing and relevant. It would be beneficial to elaborate on the specific findings of the meta-analysis or studies that support these claims.
Thank you for your valuable suggestion. We incorporated meta-analyses involving thousands of patients to elaborate the potential effects of metformin.
(Lines: 107-118)
- Line 43-50: The explanation of metformin's potential role in cancer treatment is insightful, particularly the mention of its ability to reduce circulating insulin levels and its direct inhibitory effects on cancer cells. Providing some specific examples or statistics on the success of metformin in cancer treatment would strengthen this section.
We incorporated details about the potential role of metformin in cancer treatment, along with specific examples.
(Lines: 96-106)
- Line 61-63: The statement about a consensus on other first-line treatment approaches for certain patients is relevant. If possible, provide examples or criteria for when alternative treatments might be considered.
We enhanced the discussion regarding metformin, highlighting its advantages and disadvantages. Additionally, based on your valuable input, we included a new paragraph in the summary, presenting the concept of alternative treatments tailored to a patient's specific risk factors.
(Lines: 601-677, Figure 1)
- Line 175: "Acrabose (Precose)" should be corrected to "Acarbose (Precose)".
Thank you for your comment, this has been fixed.
(Line 249)
- Line 175-177: There's an inconsistency with the dates. The FDA approved Acarbose in 1999, but the clinical trial was published in 1995. If Acarbose was only approved in 1999, how was a trial published in 1995? This needs clarification.
Thank you for your comment. Upon re-examination, the results of a multicenter double-blind placebo-controlled trial were published in 1995. Additionally, in 1999, the FDA approved the use of Acarbose.
Attached are the two references for your convenience.
Coniff RF, Shapiro JA, Robbins D, Kleinfield R, Seaton TB, Beisswenger P, McGill JB. Reduction of glycosylated hemoglobin and postprandial hyperglycemia by acarbose in patients with NIDDM. A placebo-controlled dose-comparison study. Diabetes Care. 1995 Jun;18(6):817-24. doi: 10.2337/diacare.18.6.817. PMID: 7555508.
https://www.accessdata.fda.gov/drugsatfda_docs/nda/99/020482S010_Precose.cfm
- Line 185: "incase" should be two words: "in case."
Thank you for your comment, this has been fixed
(Line 258)
- Please fix the references in the manuscript uniformly
Done. Thank you for pointing this out.
- Please reduce the similarities level from the manuscript.
We did our best to reduce similarity.
Comments on the Quality of English Language
The overall quality of English language and grammar in the draft is quite good. The sentences are well-structured, and the language used is clear and concise. However, there are a few minor grammatical issues and areas where language could be further improved for clarity.
The paper has been reviewed by a medical editor.
Reviewer 4 Report
Comments and Suggestions for Authors
The manuscript entitled “Drug Therapies for Diabetes” and authored by Sibony and colleagues reviewed the rapidly developing arena of pharmacotherapy in diabetes and the currently available anti-diabetic drugs. Authors then discuss those drugs’ effectiveness, efficacy, impact on target organs, side effects, and insights for personalized therapeutic protocols. The manuscript is well thought through and written and would be recommended for further processing if the following minor comments were addressed. What time range of publication did this review article cover, what keywords did the search for literature include, what were the inclusion criteria, how many studies did the search find and how many were primary research vs review articles, of those, how many were selected for evaluation in this study, and finally what criteria were used for selecting the articles that were reviewed (was it the subject of the study, its novelty or both). It’d also be interesting to briefly shed some light on patents relevant to T2DM therapy and prevention, if any.
Other comments
· Abbreviations should be carefully reviewed.
· It’d help to explain in the legend of fig 1 all the numbers stated there.
Comments on the Quality of English LanguageMinor editing would help.
Author Response
Reviewer 4
The manuscript entitled “Drug Therapies for Diabetes” and authored by Sibony and colleagues reviewed the rapidly developing arena of pharmacotherapy in diabetes and the currently available anti-diabetic drugs. Authors then discuss those drugs’ effectiveness, efficacy, impact on target organs, side effects, and insights for personalized therapeutic protocols.
The manuscript is well thought through and written and would be recommended for further processing if the following minor comments were addressed.
Thank you for your positive feedback. We welcome your great comments and edits below and have incorporated them carefully to improve the manuscript.
What time range of publication did this review article cover, what keywords did the search for literature include, what were the inclusion criteria, how many studies did the search find and how many were primary research vs review articles, of those, how many were selected for evaluation in this study, and finally what criteria were used for selecting the articles that were reviewed (was it the subject of the study, its novelty or both).
We reviewed literature spanning from September 2023 and backward, utilizing numerous keywords to ensure comprehensive coverage. Our approach involved examining various meta-analyses to guarantee the inclusion of the latest information in the paper. Simultaneously, we endeavored to streamline the content and references, keeping them at a reasonable and manageable quantity. Consequently, we chose not to incorporate papers that we deemed less relevant or lacking in new information. Moreover, the readers have the option to examine the referenced sources and observe the precise dates for further details.
It’d also be interesting to briefly shed some light on patents relevant to T2DM therapy and prevention, if any.
Thank you for your suggestion which could improve our paper. However, it would necessitate adding many references on the one hand and going in the details of each in the paper. We hope you will accept our decision.
Other comments
- Abbreviations should be carefully reviewed.
Thank you for your comment, they have been fixed.
- It’d help to explain in the legend of fig 1 all the numbers stated there.
We introduced a new figure and in line with your suggestion, included a description and explanation detailing its contents.
(Figure 1)
Comments on the Quality of English LanguageMinor editing would help.
The paper has been reviewed by a professional medical editor.
Reviewer 5 Report
Comments and Suggestions for Authors
Dear Authors,
The manuscript entitled "Drug Therapies for Diabetes" includes different used drug therapies since ages and new drugs. However, there is different literature regarding the same and such treatment is discussed widely.
The current compilation of all the treatments in one review is good for readers and will be helpful for researchers.
The manuscript direcly start with heading metformin without any general statement. A basic general introduction should be there before starting therapies.
The last image before the reference is not readable and the size should be enhanced for better visibility.
The manuscript may be accepted for publication after the suggested minor changes..
Author Response
Reviewer 5
Dear Authors,
The manuscript entitled "Drug Therapies for Diabetes" includes different used drug therapies since ages and new drugs. However, there is different literature regarding the same and such treatment is discussed widely.
The current compilation of all the treatments in one review is good for readers and will be helpful for researchers.
Thank you for your positive feedback. We welcome your comments and suggestions and have incorporated them to improve the manuscript.
An itemized point-by-point response to your comments is presented below.
The manuscript direcly start with heading metformin without any general statement. A basic general introduction should be there before starting therapies.
Thank you for this point. As suggested, we included an opening paragraph wherein we elaborated on the topic and described the purpose of the review.
(Lines 28 - 51)
The last image before the reference is not readable and the size should be enhanced for better visibility.
After revising the paper, we replaced the existing guideline and add a new proposal.
(Figure 1)
The manuscript may be accepted for publication after the suggested minor changes..
Thank you.
Round 2
Reviewer 1 Report
Comments and Suggestions for Authors
After corrections as per comments authors significantly improved the quality of their manuscript.